# Within-plant genetic drift to control virus adaptation to host resistance genes

**Lucie Tamisier**[1,2]☯*, **Frédéric Fabre**[3]☯, **Marion Szadkowski**[1,2], **Lola Chateau**[1], **Ghislaine Nemouchi**[2], **Grégory Girardot**[1], **Pauline Millot**[1], **Alain Palloix**[2], **Benoît Moury**[1]

**1** INRAE, Pathologie Végétale, F-84140 Montfavet, France, **2** INRAE, Génétique et Amélioration des Fruits et Légumes, F-84143 Montfavet, France, **3** INRAE, UMR SAVE, 33882 Villenave d'Ornon, France

☯ These authors contributed equally to this work.
* lucie.tamisier@inrae.fr

**Data Availability Statement:** The authors confirm that all data underlying the findings are fully available without restriction. All relevant data are

## Abstract

Manipulating evolutionary forces imposed by hosts on pathogens like genetic drift and selection could avoid the emergence of virulent pathogens. For instance, increasing genetic drift could decrease the risk of pathogen adaptation through the random fixation of deleterious mutations or the elimination of favorable ones in the pathogen population. However, no experimental proof of this approach is available for a plant-pathogen system. We studied the impact of pepper (*Capsicum annuum*) lines carrying the same major resistance gene but contrasted genetic backgrounds on the evolution of *Potato virus Y* (PVY). The pepper lines were chosen for the contrasted levels of genetic drift (inversely related to $N_e$, the effective population size) they exert on PVY populations, as well as for their contrasted resistance efficiency (inversely related to the initial replicative fitness, $W_i$, of PVY in these lines). Experimental evolution was performed by serially passaging 64 PVY populations every month on six contrasted pepper lines during seven months. These PVY populations exhibited highly divergent evolutionary trajectories, ranging from viral extinctions to replicative fitness gains. The sequencing of the PVY VPg cistron, where adaptive mutations are likely to occur, allowed linking these replicative fitness gains to parallel adaptive nonsynonymous mutations. Evolutionary trajectories were well explained by the genetic drift imposed by the host. More specifically, $N_e$, $W_i$ and their synergistic interaction played a major role in the fate of PVY populations. When $N_e$ was low (*i.e.* strong genetic drift), the final PVY replicative fitness remained close to the initial replicative fitness, whereas when $N_e$ was high (*i.e.* low genetic drift), the final PVY replicative fitness was high independently of the replicative fitness of the initially inoculated virus. We show that combining a high resistance efficiency (low $W_i$) and a strong genetic drift (low $N_e$) is the best solution to increase resistance durability, that is, to avoid virus adaptation on the long term.

## Author summary

Given their high evolutionary potential, viruses are notoriously difficult to control by drug treatments or by breeding resistant hosts. In plants, resistance genes targeting viruses

within the paper and its Supporting information files.

**Funding:** This study was supported by a PhD fellowship from the department 'Biologie et Amélioration des Plantes' (BAP) of INRAE, the Sustainable Management of Crop Health (SMaCH) INRAE metaprogramme, and the Région Provence-Alpes-Côte d'Azur (PACA) to LT, and by the ANR PRC project ArchiV (grant no. ANR-18-CE32-0004-01) obtained by BM. The funders had no role in study design, data collection and analysis, decision to publish, or preparation of the manuscript.

**Competing interests:** The authors have declared that no competing interests exist.

become often rapidly inefficient due to virus mutational escape. Manipulating the evolutionary forces imposed by the plant on the virus population, like selection and genetic drift, could help control virus adaptation and prevent resistance breakdown. Theoretically, a plant genotype imposing low selective pressures could slow down the fixation of adaptive viral mutations, while a plant genotype imposing strong genetic drift could decrease virus fitness by favoring the random fixation of deleterious viral mutations and/or the elimination of favorable ones. Using experimental evolution, we studied the effect of plants imposing contrasted levels of genetic drift on the emergence of resistance-adapted virus mutants. Results indicate that the level of genetic drift during plant colonization, in part controlled by the genetic background of the plant, and the plant resistance level are the main drivers of the virus evolutionary trajectories, which varied from fast adaptation to extinction. Consequently, it is possible to achieve higher resistance efficiency and durability by breeding plant genotypes through the introduction of major-effect resistance genes exerting a strong selection on virus populations into a genetic background increasing genetic drift.

## Introduction

The cultivation of plants carrying resistance genes is one of the favorite levers to control viral diseases in agriculture because of its efficiency, the simplicity of its implementation and the absence of harmful impacts on the environment and human health. However, the deployment of such plants is frequently followed by a 'resistance breakdown', *i.e.* a loss of resistance efficiency due to virus adaptation [1,2]. Indeed, the strong directional selection exerted by the resistant plants onto virus population favors their rapid adaptation. Reducing the intensity of this selection by using partial instead of high-level resistance has been proposed to avoid resistance breakdown [3–5]. Selection is a deterministic evolutionary force that increases the frequency of the fittest virus variants over time. Its strength is usually measured with the selection coefficient *s*, defined as the difference in fitness between two virus variants. However, this is not the sole action lever for breeders. Acting on genetic drift, an evolutionary force known to counteract the effect of selection, is also an option.

Genetic drift is a stochastic force inducing random variations of viral variants frequencies from generation to generation. Its strength depends on the effective population size ($N_e$), defined as the number of viral variants that effectively transmit their genes to the next generation [6]: the lower $N_e$, the stronger the genetic drift. Since $N_e$ is a key parameter to understand the effect of genetic drift on virus evolution, many studies have focused on the assessment of $N_e$ of plant RNA viruses. Most of these studies found very small $N_e$ values, usually much lower than the census size of the population [7]. The main reason is that the viral population size varies considerably during the infection process due to the action of repetitive bottlenecks during inoculation, within-host colonization and between-host transmission [8], which strongly decrease $N_e$.

The probability of fixation of a new mutation in a viral population depends both on $N_e$ and *s* [9]. A genetic drift regime occurs when $N_e \times |s| \ll 1$. In this case, genetic drift predominates over selection and the probabilities of fixation of favorable and deleterious mutations become similar to those of neutral mutations [10]. Some favorable mutations could be lost, while some deleterious mutations could be fixed and potentially cause a loss of fitness by accumulating in the population. Conversely, a selection regime occurs when $N_e \times |s| \gg 1$. In this case, selection prevails over genetic drift and the probabilities of fixation of mutations will depend mostly on *s*,

the most favorable mutations being fixed and deleterious mutations being eliminated. Therefore, one may theoretically take advantage of plant cultivars characterized by low pathogen $N_e$ to increase the fixation of deleterious mutations and/or decrease the fixation of favorable ones. To test these hypotheses on virus adaptability, we used pepper (*Capsicum annuum*, family Solanaceae) doubled-haploid (DH) lines shown to impose contrasted evolutionary regimes to *Potato virus Y* (PVY; genus *Potyvirus*, family *Potyviridae*) populations by Rousseau et al [11].

Here, we examined how six pepper genotypes of the same plant species carrying the same major-effect resistance gene but imposing different intensities of genetic drift onto PVY populations impacted virus evolutionary trajectory. We performed an evolution experiment through the serial passages of sixty-four PVY populations every month on six contrasted pepper DH lines during seven months. Our results demonstrated that combining a strong genetic drift and a low initial PVY fitness can prevent virus adaptation. From an agronomic point of view, selecting cultivars based on the levels of selection and genetic drift they exert on a composite virus population could help us to take control of virus evolution by manipulating the evolutionary forces acting on virus populations within the plant.

## Results and discussion

### Diverse evolutionary trajectories among PVY lineages: Extinction, status quo or replicative fitness gains associated with parallel fixation of mutations

In order to study the effects of different evolutionary constraints corresponding to different host plant environments on virus evolution, we conducted an experimental evolution of PVY on closely-related pepper DH lines derived from the same $F_1$ hybrid between two parental lines and carrying the PVY major-effect resistance allele *pvr2³*. Sixty-four independent PVY evolutionary lineages were obtained by serial passages in six DH lines contrasted for $N_e$ and the viral load of the PVY clones that started the experiment ($W_i$) (Fig 1 and S1 Table). Viral load is one component of the viral fitness, together with host range and transmissibility. In the following we will refer to this fitness component as 'replicative fitness' following Wargo and Kurath [12] and Elena [13], and we will denote it by $W$.

Three PVY variants derived from infectious cDNA clones were used. They were named SON41-101G, SON41-119N and SON41-115K according to their amino acid position and type in the VPg (viral protein genome-linked) that distinguish them from PVY isolate SON41p. These three mutations were shown to be responsible for PVY adaptation to the pepper resistance allele *pvr2³* [14] and to confer respectively a low, medium and high level of adaptation to the resistance allele [11]. The PVY SON41-119N cDNA clone was chosen to start the evolution experiment for all DH lines, its intermediate level of adaptation to *pvr2³* guaranteeing successful infection while allowing the virus to adapt further during the evolution experiment. The SON41-101G and -115K clones were passaged only on DH line HD2173 (Fig 2), in order to compare virus variants with different levels of initial adaptation on the same host. After each passage, the VPg cistron of the PVY populations, where adaptive mutations are likely to occur [14–19], was sequenced and single nucleotide polymorphisms (SNPs) were detected by comparison with the sequence of the starting clones.

Among the 64 PVY lineages, nine went to extinction as none of the inoculated plants was infected after two to four infection cycles (Fig 3). Three lineages out of eight became extinct in pepper line HD2256 and six of eight in line HD2321. Then, 32 lineages did not show any mutation in the VPg cistron, where most mutations responsible for PVY adaptation to *pvr2³* were likely to occur [14,20]. Most of these correspond to pepper lines HD2256, HD219 and HD2173 with PVY variant SON41-119N, and pepper line HD2173 with PVY variant SON41-

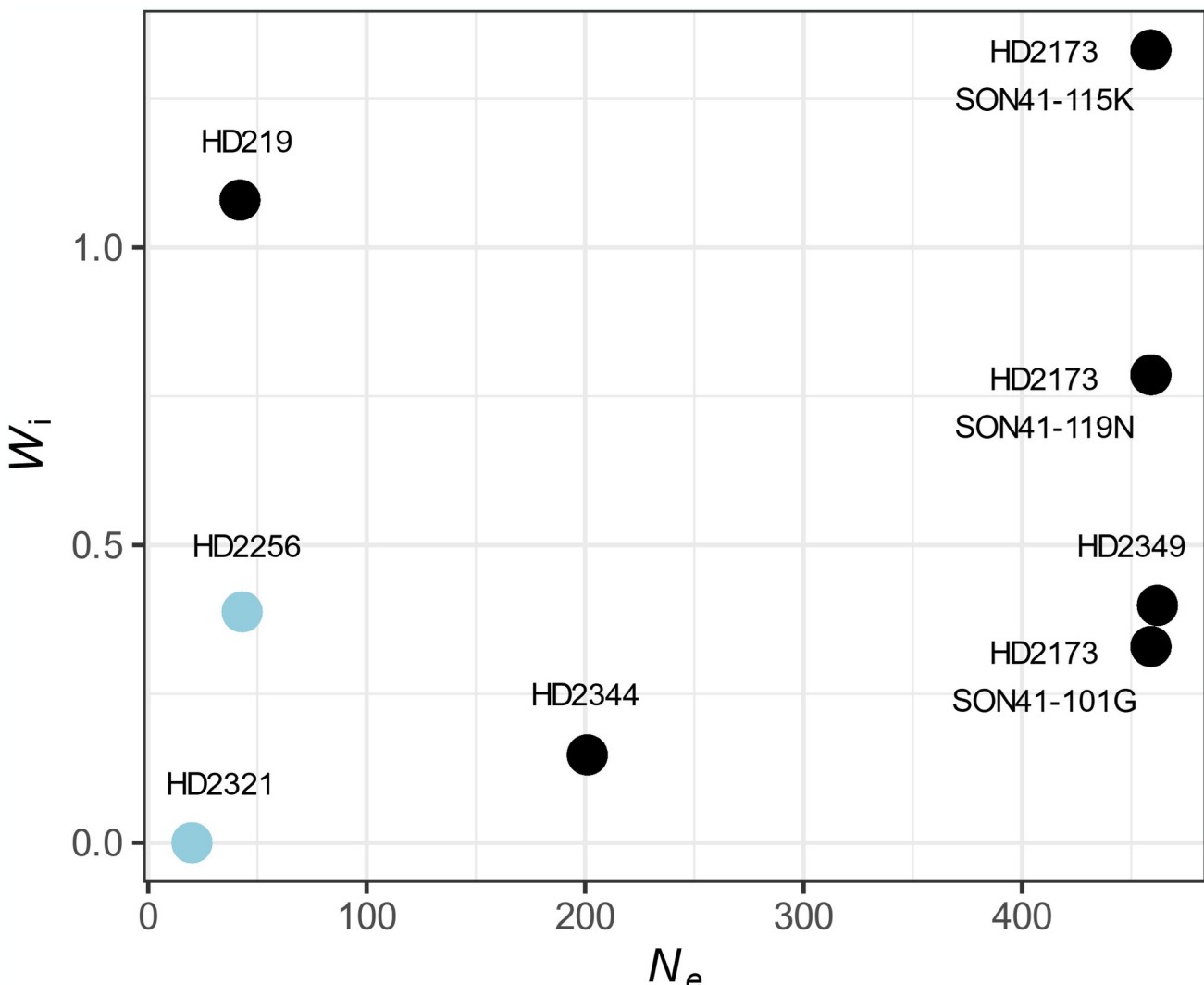

**Fig 1. Features of the pepper DH line—Initial PVY variant combinations used for the evolution experiment.** The pepper lines HD219, HD2256, HD2321, HD2349 and HD2344 were inoculated only with the PVY SON41-119N variant, while HD2173 was also separately inoculated with the SON41-115K and SON41-101G variants. The eight virus-host combinations are plotted according to two traits: (i) the effective population size ($N_e$) and (ii) the initial virus replicative fitness ($W_i$) of each variant. $N_e$ was estimated from 7 to 10 dpi by Rousseau et al [11], using a PVY population composed of an equimolar mixture of five PVY variants, including the three used in this study (see Materials and methods). $W_i$ was measured in the present study. The two DH line—Initial PVY variant combinations with both low $W_i$ and $N_e$ levels are highlighted in blue.

115K. Finally, 24 lineages have shown at least one *de novo* nucleotide substitution in the VPg cistron, including one lineage (L22) that went extinct. A total of 30 substitutions, 27 nonsynonymous and three synonymous, occurred independently in those lineages. All of them were either fixed in the population as soon as their first detection or became fixed during the next infection cycle, with only two exceptions (Fig 3). Overall, among the seven different nonsynonymous substitutions observed, four arose in several independent lineages (mutations 102K, 115K, 115M and 119N). Three of the PVY populations remaining at the end of the experiment were VPg double mutants. Regarding the 48 lineages deriving from PVY variant SON41-119N, the most frequent nonsynonymous mutation in the VPg cistron was 115M (eight lineages) followed by 115K (five lineages). In addition, six of the eight lineages coming from PVY

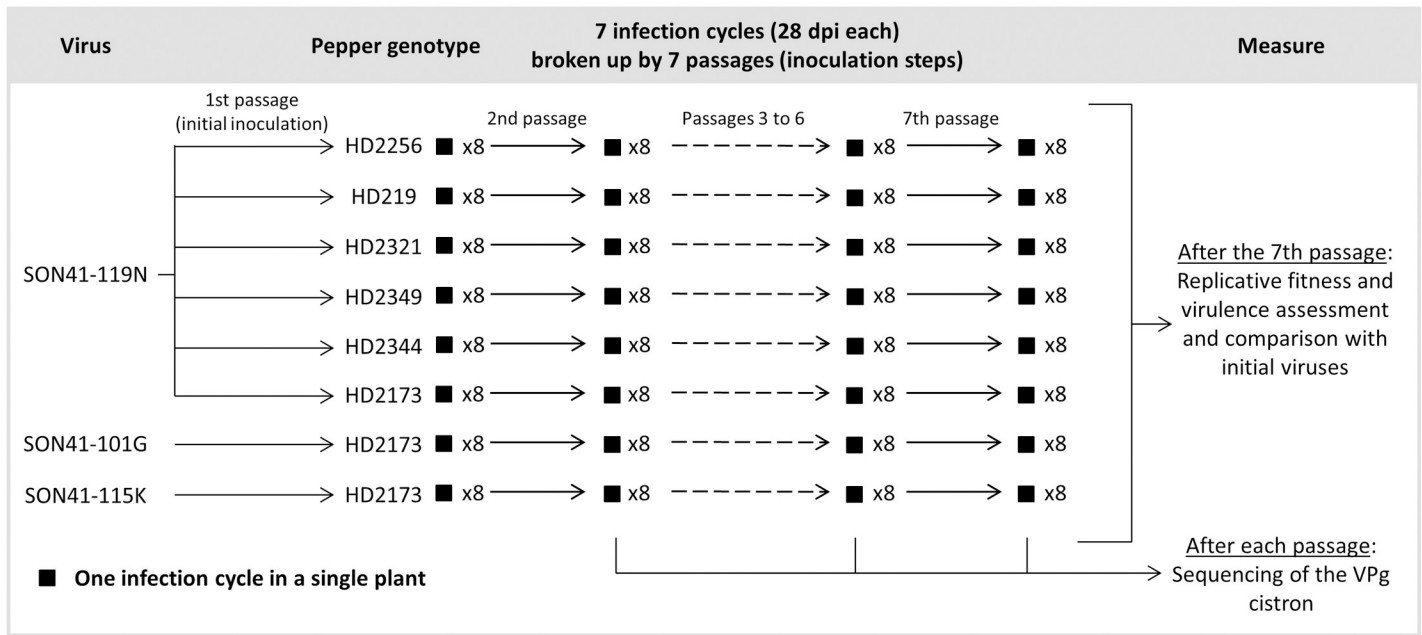

**Fig 2. Experimental design of the study.** Six pepper genotypes (doubled-haploid lines) were inoculated with the SON41-119N variant of PVY. The HD2173 line was also inoculated with either the SON41-101G or SON41-115K variants. For each virus-host combination, eight PVY evolutionary lineages, *i.e.* temporal series of virus populations, connected by a continuous line of descent in the same pepper genotype, were propagated during seven infection cycles of 28 days. The VPg cistron of the 64 evolutionary lineages was sequenced at the end of each infection cycle. Eventually, the replicative fitness and virulence of the final PVY populations were assessed.

variant SON41-101G fixed mutation 119N. Finally, no substitution was detected in the eight lineages derived from PVY variant SON41-115K.

After seven infection cycles, the replicative fitness of the initial PVY variants and final PVY populations ($W_i$ and $W_f$, respectively) were compared in the pepper line where each PVY population had evolved in an independent experiment (S1 Table and S1 Dataset). The change in virus replicative fitness during the experiment was measured as $\Delta W = W_f - W_i$. A significant replicative fitness increase ($\Delta W > 0$) was observed globally for the populations derived from PVY variant SON41-119N serially inoculated in the three pepper lines HD2321, HD2349 and HD2344 and for the populations derived from PVY variant SON41-101G serially inoculated in line HD2173 (Wilcoxon tests, $p < 0.001$, Fig 4A). More precisely, $\Delta W$ was significantly positive for the two remaining populations in HD2321, for three of eight populations in HD2349, three of eight populations in HD2344 and all populations in HD2173 (Dunnett tests, Fig 4B, 4C, 4D and 4E). Replicative fitness gains were almost always associated with the fixation of one or two nonsynonymous substitutions in the VPg cistron (15 of 16 PVY populations). Most frequently, mutations 119N (six cases), 115K (five cases) and 115M (four cases) were associated with significant replicative fitness increases. Even though mutations in the VPg are correlated with the observed fitness changes, it should be noted that other mutations elsewhere in unsequenced regions of the genome could also impact PVY replicative fitness [20–22].

In contrast with these replicative fitness changes, little change in PVY virulence (*i.e.* damage caused by the virus to the plant) was measured based on the decrease of plant fresh weight and height (S1 and S2 Figs). However, this result does not mean that the virus adaptation does not incur any additional costs for the plant. Other plant health traits that we did not measure (leaf chlorophyll content, number of fruits, size and weight of older plants. . .) may be affected by virus adaptation.

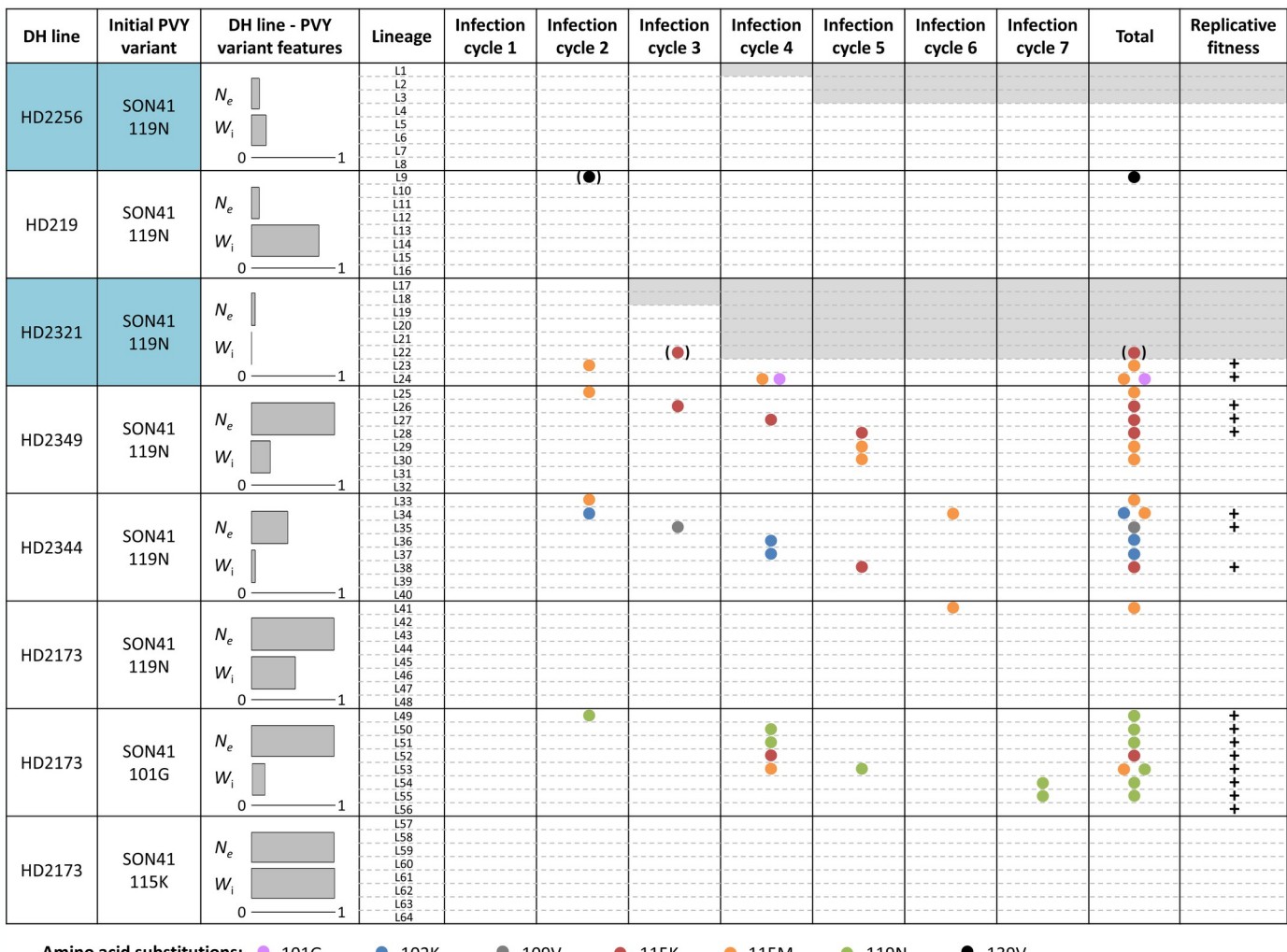

**Fig 3. Amino acid substitutions detected for the 64 *Potato virus Y* (PVY) lineages during seven infection cycles.** The effective population size ($N_e$) estimated from 7 to 10 dpi in Rousseau et al [11] (see Materials and Methods) and the initial virus replicative fitness ($W_i$) characterizing each pepper line–PVY variant combination are expressed as a fraction of their maximum value. Each row represents a virus lineage, named L1 to L64. Substitutions are represented by dots plotted according to their order or appearance and the colors associated to the dot distinguish the different substitutions. A grey rectangle indicates the extinction of a lineage, *i.e.* the absence of PVY detection in plants newly inoculated at the end of the last infection cycle. Substitutions in brackets were detected but not fixed in the population (mutation 139V in lineage L9 was detected at the end of cycle 2 but was not fixed until cycle 6 and mutation 115K in lineage L22 was detected at the end of cycle 3 but the virus population get extinct during the subsequent infection cycle). The plus signs indicate a significant increase of PVY replicative fitness after seven infection cycles compared to the initial PVY replicative fitness. The two DH line—Initial PVY variant combinations with both low $W_i$ and $N_e$ levels are highlighted in blue.

VPg mutations 101G, 115K and 119N, corresponding altogether to 12 of 16 cases of replicative fitness gains, have been previously shown to determine adaptation to the *pvr2³* major-effect resistance gene present in all pepper lines used in this study [14,23]. We validated by reverse genetics that the two substitutions 115K and 115M arising most frequently in lineages derived from SON41-119N were responsible for PVY replicative fitness gains in the pepper lines where they have appeared (HD2173, HD2321, HD2344 and HD2349) but also in the other pepper lines (both substitutions in line HD219 and substitution 115K in line HD2256) (S2 and S3 Tables). Consequently, the fact that none of the PVY lineages in HD219 or HD2256 and only two of eight lineages in HD2321 showed an increase in replicative fitness cannot be

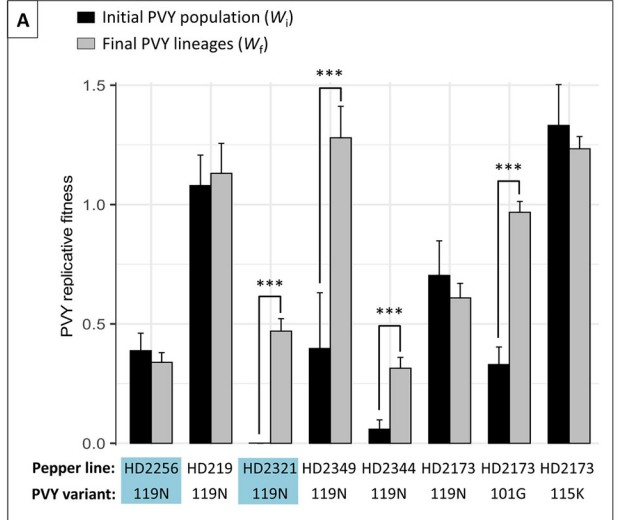

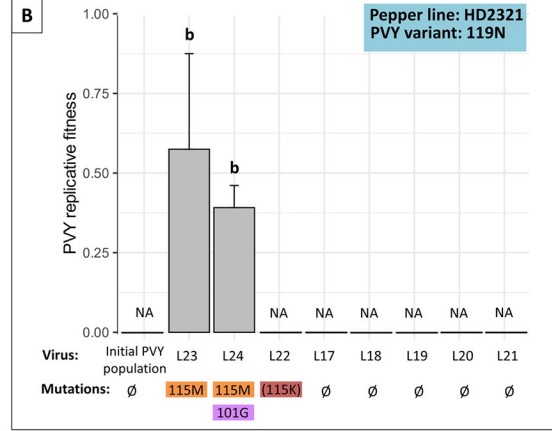

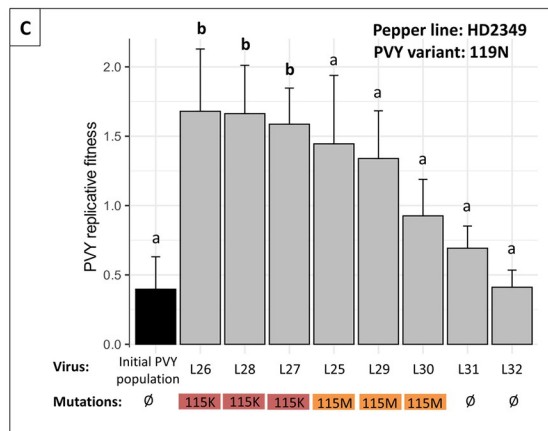

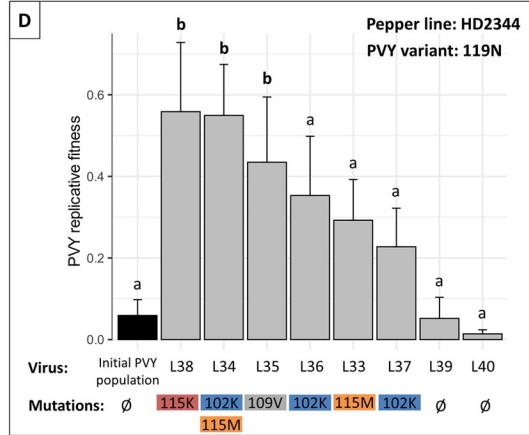

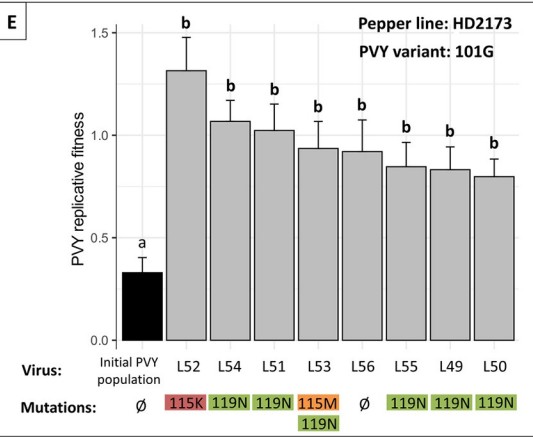

**Fig 4. Replicative fitness changes during the evolution experiment.** Replicative fitness of the initial PVY variant ($W_i$) (black) and of the final populations that did not go extinct ($W_f$) (grey) in six pepper lines (A). The PVY variants used to start the infection are variants of the SON41p clone. For each pepper DH line, stars represent significant replicative fitness differences between the initial and final PVY populations (Wilcoxon test, p < 0.001). Error bars represent standard errors. The four other graphics show the replicative fitness of the initial PVY variants (black) and the eight derived final populations (grey) in four pepper lines: HD2321 (B), HD2349 (C) and HD2344 (D) inoculated with SON41-119N and HD2173 inoculated with SON41-101G (E). The nonsynonymous substitutions detected in the VPg cistron of each final PVY population or the absence of detected mutation (Ø) are indicated. The substitution in bracket was not fixed in the lineage. The letters a and b represent the different groups obtained after the comparison of each final population to the initial PVY variant using Dunnett test (p < 0.05). NA: not measured (extinct lineage). The two DH line—Initial PVY variant combinations with both low $W_i$ and $N_e$ levels are highlighted in blue. Error bars indicate standard errors.

solely attributed to the lack of adaptive mutations in the VPg to these plant genotypes. Instead, it is more likely due to the evolutionary forces exerted by the plant genotypes onto the virus population that prevented their fixation. Note that a low mutation supply could also explain the absence of adaptive mutations in HD2256. Assuming that the number of mutations correlates with the intra-plant viral concentration ($W_i$), it is possible that the mutation supply is lower in HD2256, which has a low $W_i$ (unlike HD219). The impact of evolutionary forces combined with this low mutation supply could explain the results for this DH line.

## Relevance and accuracy of variables used to explain the virus evolutionary trajectories

Pepper genotype had a significant effect on the number of VPg mutations that were fixed in the PVY populations (Pearson's chi-squared test, $\chi 2 = 35.24$, $p < 0.01$) and on change in PVY replicative fitness $\Delta W$ (one-way ANOVA, F = 13.68, $p < 0.001$). These results are probably due to the evolutionary forces that the pepper genotypes exert on PVY populations. To test this hypothesis, we analysed if several variables linked to these evolutionary forces could explain the PVY evolutionary trajectories. We previously estimated $N_e$ of the whole viral population at different stages of the infection [11], using an indirect estimation method based on the change of the relative frequencies of virus variants over time. We also estimated $N_e$ at the inoculation step for the three PVY variants used to initiate the evolution experiment (SON41-101G, SON41-115K and SON41-119N), using a direct estimation method following the demographic changes occurring in the viral population.

Briefly, the dynamics of an artificial PVY population composed of an equimolar mixture of five SON41p VPg variants carrying one or two non-synonymous substitutions (SON41-101G, -115K, -119N, -101G-115K and -115K-119N variants) was followed during plant infection [11]. The same pepper DH lines as the ones used in the present study were inoculated with this artificial PVY population. The relative frequencies of the different PVY variants were precisely estimated at several timepoints post-inoculation by sequencing (MiSeq Illumina) the region of the VPg cistron where the mutations distinguishing these variants were located. The effective population size ($N_e$) of the PVY population from 0 to 34 days post inoculation (dpi) was estimated using a multi-allelic Wright-Fisher model. Multiple estimates of $N_e$ were available from this previous experiment: $N_e$ estimated from 1 to 6 dpi, 7 to 10 dpi, 11 to 14 dpi, 15 to 34 dpi as well as the harmonic mean of all $N_e$ estimates (S2 Dataset). Regarding $N_e$ at the inoculation step, the number of infection foci on the cotyledons initiated by three PVY infectious clones (SON41-101G, SON41-115K and SON41-119N) expressing the green fluorescent protein (GFP) were counted for each DH line—PVY-GFP variant combination. We previously demonstrated that each primary infection focus was initiated by a single PVY particle [24], allowing to estimate $N_e$ at the inoculation step (see the Materials and Methods section for more details) (S2 Dataset).

We investigated the effects of $W_i$ and $N_e$ on (i) the probability that a PVY lineage experiences a replicative fitness gain and on (ii) the PVY final replicative fitness $W_f$. Generalized linear models (GLMs) were used to study the effects of $W_i$ and $N_e$ on these variables. Due to multicollinearity among the explanatory variables, we were unable to include all seven explanatory variables in the models (comprising six $N_e$ estimates and $W_i$). Specifically, $N_e$ estimates at the inoculation step, as well as those estimated for the 7 to 10 dpi, 11 to 14 dpi, 15 to 34 dpi intervals and the harmonic mean of all $N_e$ estimates, exhibited positive and significant correlations (Spearman's coefficient ranging between 0.83 and 0.39, $p < 0.01$), while $N_e$ estimated from 1 to 6 dpi and $W_i$ showed a negative and significant correlation (Spearman's coefficient = $-0.55$, $p < 0.001$) (S1 and S2 Files). Moreover, the 35 models incorporating all possible

combinations of three explanatory variables (two $N_e$ estimates and $W_i$, or three $N_e$ estimates) displayed extremely high Variance Inflation Factors (VIFs) exceeding 100 (S1 and S2 Files). Among the 21 models comprising all possible combinations of two explanatory variables (one $N_e$ estimate and $W_i$, or two $N_e$ estimates), several exhibited VIFs lower than 10. The Akaike Information Criterion (AIC) was used to select the best model. The models exhibiting the lowest AIC always included $W_i$ and $N_e$ estimated from 7 to 10 dpi, for both response variables (S1 and S2 Files). Moreover, McFadden's R-squared and R-squared was utilized to assess model fit, for the probability of replicative fitness gain and the PVY final replicative fitness $W_f$, respectively. Again, the models exhibiting the highest McFadden's R-squared and R-squared always included $W_i$ and $N_e$ estimated from 7 to 10 dpi, for both response variables (McFadden's R-squared = 0.33 and R-squared = 0.42, for the probability of replicative fitness gain and the PVY final replicative fitness $W_f$, respectively; Table 1 and S1 and S2 Files). Therefore, only $N_e$ estimated from 7 to 10 dpi was finally retained in the GLMs.

We can assume that since $N_e$ estimates the intensity of genetic drift, which is a stochastic evolutionary force, it is weakly impacted by the composition of the viral population. Therefore, even if $N_e$ estimated from 7 to 10 dpi was estimated in an experimental context that is not identical to ours, it could still be a relevant parameter to explain our data. It should be mentioned that $N_e$ value could vary along time as the composition of the viral population changes. Our measurements may not be representative of all the values this variable can take. Nevertheless, $N_e$ estimated from 7 to 10 dpi was precisely estimated with a quite high heritability ($h^2 = 0.63$) [11]. We will call this estimate $N_e$ in the following of this article.

Although our GLMs explained much of the variation in PVY adaptation, additional explanatory variables could be estimated to further explain PVY adaptation. Among these variables, plant tolerance to virus infection could significantly influence virus adaptation to host resistance, as shown by Montes et al [25]. The pepper DH lines used in the present study showed varying degrees of symptom, which could be related to different levels of tolerance. However, we did not carry out a thorough quantitative estimation of these tolerance differences, which is necessary to incorporate them into GLMs.

**Table 1. Generalized linear models of the probability of PVY replicative fitness gain during evolution experiment and of PVY final replicative fitness.** A logistic regression was used to analyze the effects of the initial replicative fitness $W_i$, the effective population size at the onset of systemic infection $N_e$ and their interaction on the probability that a PVY lineage shows a replicative fitness gain. The analysis was conducted on the 64 lineages of the experiment. Linear regression was used to investigate the effects of these same explanatory variables on the final replicative fitness $W_f$, using the 55 lineages (out of 64) not extinct at the end of the experiment.

| Explanatory variable | Estimate | Std. Error | z or t value [1] | p-value |
|---|---|---|---|---|
| **Probability of replicative fitness gain ($n = 64$)** | | | | |
| Intercept | −1.640 | 0.737 | −2.225 | 0.026 |
| $W_i$ | 1.589 | 1.075 | 1.478 | 0.139 |
| $N_e$ | 0.015 | 0.004 | 3.741 | $1.8 \times 10^{-4}$ |
| $W_i \times N_e$ | −0.018 | 0.005 | −3.291 | 0.001 |
| McFadden's R-squared: 0.33 | | | | |
| **Final replicative fitness $W_f$ ($n = 55$)** | | | | |
| Intercept | −0.019 | 0.160 | −0.122 | 0.903 |
| $W_i$ | 1.033 | 0.213 | 4.858 | $1.2 \times 10^{-5}$ |
| $N_e$ | 0.002 | $4.9 \times 10^{-4}$ | 3.980 | $2.2 \times 10^{-4}$ |
| $W_i \times N_e$ | −0.002 | $6.2 \times 10^{-4}$ | −3.108 | 0.003 |
| R-squared: 0.42 | | | | |

[1] z-value are reported for logistic regression while t-values are reported for linear regressions

## Combining a strong within-plant genetic drift and a low PVY initial replicative fitness can prevent virus adaptation to host-resistance genes

We first used logistic regression to analyze the effects of $W_i$, $N_e$, and their interaction on the probability that a PVY lineage experiences a replicative fitness gain. After a stepwise model selection performed on the full model using the Akaike Information Criterion (AIC), the model including $W_i$, $N_e$ and their interaction was retained. Both $N_e$ (p-value = $1.8\times10^{-4}$) and its interaction with $W_i$ (p-value = $1\times10^{-3}$) were significant (Table 1). Similar results were obtained after removing the nine lineages going to extinction before the end of the experiment (S1 and S2 Files). Overall, the probability of replicative fitness gain increases with decreasing $W_i$ for high $N_e$ (*i.e.* broad bottleneck). This relationship tends to reverse for low $N_e$ (*i.e.* narrow bottleneck) (Fig 5A).

Then, we performed linear regressions to investigate the effects of $W_i$, $N_e$, and their interaction on the final replicative fitness $W_f$ measured on the 55 lineages that did not go to extinction during the experiment. As previously, the stepwise selection model retained $W_i$ (p-value = $1.2\times10^{-5}$), $N_e$ (p-value = $2.2\times10^{-4}$) and their interaction (p-value = $3.0\times10^{-3}$) (Table 1). Overall, the final replicative fitness $W_f$ remains close to the initial replicative fitness when a narrow bottleneck occurs at the onset of systemic infection (Fig 5B). Things go differently in plant genotypes characterized by high $N_e$ (*i.e.* broad bottleneck). In these genotypes, final replicative fitness remains high independently of the replicative fitness of the initially inoculated virus.

In agreement with Couce and Tenaillon [25], our evolution experiment confirmed that low replicative fitness microbe genotypes adapt faster than their fitter counterparts in the absence

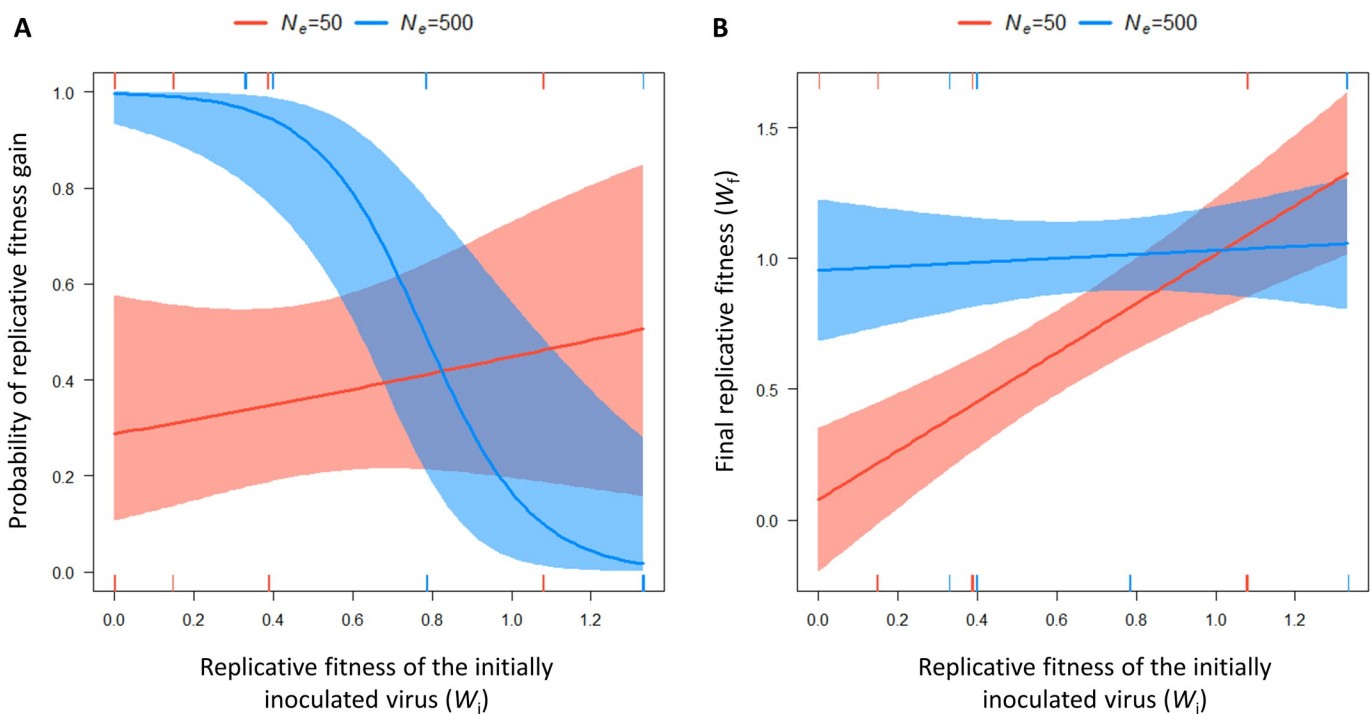

**Fig 5. Responses of the probability of replicative fitness gain (A) and final replicative fitness ($W_f$) (B) to the initial virus replicative fitness ($W_i$) and the bottleneck size at the onset of systemic infection ($N_e$).** The models are fitted on the 64 lineages for the probability of replicative fitness gain, and on the 55 lineages that did not get extinct during evolution experiment for the final replicative fitness ($W_f$). The corresponding models, explanatory variables and estimates are reported in Table 1. Rug plot are added to show the distribution of the observed data points along the x-axis.

of bottleneck (i.e., high $N_e$). In their article, Couce and Tenaillon [26] compared the replicative fitness gain, measured as the ratio $W_f/W_i$, and the initial replicative fitness $W_i$. In our models, similar results were obtained when using the replicative fitness ratio $W_f/W_i$, the replicative fitness difference $W_f-W_i$ or the final replicative fitness $W_f$ as response variables and the initial replicative fitness $W_i$ as explanatory variable. However, we chose to use $W_f$ as a response variable rather than the ratio $W_f/W_i$ or the difference $W_f-W_i$ to avoid any conundrum and misinterpretation due to "spurious correlation" between $W_f/W_i$ (or $W_f-W_i$) and $W_i$ used as an explanatory variable [27].

The observed increase of probability of replicative fitness gain for initially low replicative fitness microbes when $N_e$ was high could have several explanations linked with selection, as selection predominates over genetic drift in this situation. For instance, genotypes with a low replicative fitness could fix beneficial mutations with larger effects than genotypes with high initial replicative fitness [28]. The ratio between the numbers of beneficial and deleterious mutations available could also be different between the high replicative fitness and low replicative fitness genotypes [29]. Finally, the speed of fixation of beneficial mutations could be higher in low replicative fitness genotypes [30].

Our experiment also demonstrated that the adaptation of microbes with low initial replicative fitness can be prevented in the presence of narrow bottlenecks (small $N_e$) (Fig 5). Several explanations can be provided. First, the probability that adaptive *de novo* mutations arise in the virus population decreases with decreasing initial replicative fitness $W_i$, because of lower census population size and lower number of viral replication events. A study on the efficiency of microRNAs expressed in transgenic *Arabidopsis thaliana* plants to confer resistance to *Turnip mosaic virus* (TuMV) gave similar results. In plants expressing a suboptimal level of microRNA (increasing the virus census population size), viral populations accumulated adaptive mutations at high frequency, while the opposite was observed in plants expressing a higher level of microRNA [31]. Second, when $N_e$ is small, there is a higher chance that genetic drift will randomly eliminate some of these *de novo* mutations independently of their replicative fitness effects. Therefore, a synergistic effect between low $N_e$ and low $W_i$ could contribute to avoiding virus adaptation by decreasing the pool of adaptive mutations in the virus population and accelerating their elimination by genetic drift. Moreover, nine PVY populations evolving on pepper lines with low $N_e$ and low $W_i$ went to extinction during the experiment (Fig 3). These extinctions could be explained by the action of Muller's ratchet, a process occurring in populations of small size. Theory predicts that the population will accumulate slightly deleterious mutations over time. Individuals without deleterious mutations will become rare and will be definitively lost by genetic drift. With a continuous accumulation of deleterious mutations the population can get extinct [32]. It should also be noted that counter-examples do exist. For instance, in a rugged fitness landscape, the fixation of mutations through genetic drift can help a microbe to cross an adaptive valley from a medium fitness peak to a higher fitness peak, even though the microbe fitness in transiently lower. The impact of genetic drift on the evolution of microbial populations can therefore vary from one system to another [7].

We have shown previously that the effective size of a virus population during plant infection is a highly variable and heritable plant trait [11]. We show here that highly contrasted evolutionary trajectories, ranging from extinction to fast adaptation, affect PVY during a few infection cycles in plants characterized by contrasted virus $N_e$ and that a large part of the variation in these trajectories is attributable to an interaction effect between $N_e$ and $W_i$. The effect of $N_e$ on virus adaptation is consistent with theoretical expectations. When $N_e$ decreases, adaptive mutations have a greater chance of disappearing through genetic drift and the risk of extinction of PVY lineages increases, especially when $W_i$ is low, due to Muller's ratchet effects. Strategies potentializing genetic drift effects have also been proposed in medicine to limit the

emergence of antibiotic-resistant bacteria [33]. Two independent evolution experiments demonstrated that bottleneck sizes influence the persistence of *Escherichia coli* upon antibiotic treatment and the evolution of *E. coli* antibiotic resistance. In the first experiment, narrow bottlenecks limited the adaptive potential of *E. coli* by restricting access to highly beneficial mutations [34]. In the second one, broad bottlenecks selected for high-fitness variants showing mutations in drug target genes, while narrow bottlenecks selected for lower-fitness variants showing mutations in diverse gene classes [35]. Experimental evolution of *Pseudomonas aeruginosa* under different regimes of bottleneck sizes and selection also revealed the joint influence of these factors on the evolution of antibiotic resistance [36]. While a combination of severe bottlenecks and low antibiotic selection favored the appearance of resistance by allowing bacterial populations to recover from the bottleneck and grow large populations, the combination of severe bottlenecks and strong antibiotic selection constrained population size post-bottleneck. This decreased the probability of beneficial mutations appearing and, in some cases, led to population extinctions.

To date, the use of within-host genetic drift to prevent plant pathogen adaptation has received little attention. Since plant immunity decreases, by definition, the census size of pathogen populations and potentially also their effective size, exploiting genetic drift effects to promote the durability of plant resistance is a promising avenue of research and development of plant cultivars displaying resistance to many different kinds of pathogens and pests. Our results pave the way for new plant breeding strategies combining effective and long-lasting resistance by selecting plant genotypes imposing strong genetic drift to pathogen populations. For instance, pyramiding major resistance gene(s) with QTL(s) inducing a strong genetic drift within a plant variety would prevent virus adaptation and enhance durability.

However, the durability of plant resistance does not depend solely on the resistance genes used, but on a combination of factors, ranging from the resistance genes introgressed into the cultivars to the deployment strategies of these resistance genes in the field or in the agricultural landscape. While assessing the effect of genetic drift on the durability of plant resistance at field scale is laborious, models are powerful tools for simulating epidemics and testing various resistance deployment strategies in various conditions (presence of resistance-adapted genotype prior to deployment, effect of wild host plants on virus evolution . . .). Therefore, developing models that take into account the level of genetic drift imposed by plants on pathogens at large spatiotemporal scales is also a promising direction for the future [37].

## Materials and methods

### Virus and plant material

Three PVY (genus *Potyvirus*, family *Potyviridae*) variants derived from infectious cDNA clones were used. They were named SON41-101G, SON41-115K and SON41-119N according to their amino acid substitution (position and substituted amino acid) in the VPg (viral protein genome-linked) compared to SON41p. To ensure a high inoculum concentration, the PVY cDNA clones were first propagated in *Nicotiana tabacum* cv. Xanthi plants before inoculation of pepper plants and the concentrations of the different inocula were assessed by quantitative DAS-ELISA (Double Antibody Sandwich—Enzyme Linked Immunosorbent Assay) and adjusted when necessary. All experiments were performed in a climate-controlled growth room (20–22°C, 12-h light/day).

Six perfectly homozygous DH lines of pepper (*Capsicum annuum*, family Solanaceae) were used: HD219, HD2173, HD2256, HD2321, HD2344 and HD2349. They all derived from a cross between Yolo Wonder, an inbred line susceptible to PVY, and Perennial, an inbred line carrying the PVY resistance allele *pvr2*[3]. All DH lines carried *pvr2*[3] and differed in their genetic

background. They have been chosen based on the different levels of genetic drift that they imposed on PVY populations in previous experiments [11] (Fig 1 and S1 Table). The initial within-plant replicative fitness of PVY variants (*i.e.* the capacity of virus accumulation at the plant systemic level) were also contrasted between the DH lines. Genetic drift levels were quantified thanks to two measures of the effective population size ($N_e$). First, $N_e$ characterizing the changes in PVY variant frequencies in a composite PVY population during plant infection was estimated by Rousseau et al [11]. They have inoculated the first leaf of 15 pepper DH lines with an equimolar mixture of five PVY variants, including the three variants used in our study SON41-101G, SON41-115K and SON41-119N, as well as two double mutants named SON41-101G-115K and SON41-115K-119N. The virus populations were sampled at different time points after inoculation (6, 10, 14 and 34 dpi). From these samples, RNA extraction, reverse transcription-polymerase chain reaction (RT-PCR) and next generation sequencing (Miseq Illumina) were performed allowing to estimate the frequencies of the PVY variants in the plants. Thanks to these measures, the effective population sizes $N_e$ of the PVY population in each DH line and for each date interval have been estimated with a mechanistic-statistical model. The harmonic mean of the effective population sizes for the different date intervals was also calculated. Finally, $N_e$ estimated from 7 to 10 dpi was chosen for our analysis (see "Relevance and accuracy of parameters used to explain the virus evolutionary trajectories" section for more details). Second, $N_e$ at the inoculation step was measured during another experiment. We used three cDNA clones of PVY isolate SON41p carrying either the 101G, 115K or 119N substitution in the VPg. All clones were also tagged with the green fluorescent protein (GFP) reporter gene. The three PVY-GFP cDNA clones were first inoculated in *N. clevelandii* plants. The concentrations of the three inocula were assessed by quantitative DAS-ELISA and adjusted when necessary. Then, each inoculum was used to mechanically inoculate 10 plants on their two cotyledons for each pepper DH line, 3 weeks after sowing. At 5 and 7 dpi, the number of primary infection foci on each inoculated cotyledon was counted under a specific light wavelength (450–490 nm). We previously demonstrated that each primary infection focus was initiated by a single PVY particle [24]. Therefore, the number of primary infection foci allows to estimate $N_e$ at the inoculation step. The initial replicative fitness ($W_i$) was assessed in all DH lines for SON41-119N and only in HD2173 for SON41-101G and SON41-115K during this study. Pepper plants were mechanically inoculated on their two cotyledons, with 20 plants per DH line. At 30 dpi, a quantitative DAS-ELISA was performed as described by Ayme et al [14]. The mean virus $W_i$ in each DH line was assessed using serial dilutions of the infected plants and calculated relatively to a common control sample added in each ELISA plate.

## Experimental evolution

Sixteen plants per DH line were inoculated with PVY variant SON41-119N on their two cotyledons in order to initiate 8 independent evolutionary lineages for each DH line. DH line HD2173 was also inoculated with variants SON41-101G and SON41-115K (Fig 2). Twenty-eight days post inoculation, 8 plants of the 16 were randomly chosen for each DH line-PVY variant combination to initiate the 64 PVY lineages. For each of these plants, 1 gram from 3 apical leaves showing mosaic symptoms was ground in a phosphate buffer (0.03 M $Na_2HP0_4$, 0.2% sodium diethyldithiocarbamate, 4 mL buffer per gram of leaves) and used to inoculate 3 plants of the same genotype per evolutionary lineage. Then, one of the three plants was randomly selected to inoculate the three plants of the next infection cycle. A total of seven infection cycles of 28 days were performed similarly.

## Virus sequencing

At the end of each cycle, a sample of 200 μl was removed from each of the 64 inocula and kept at −20˚C. From these samples, we performed RNA extraction using Trizol and RT-PCR of the VPg cistron using specific primers (Forward primer: 5'-GACCTTAAGCTGAAGG-GAGTTTGGAAGAAGTCGC-3'; Reverse primer: 5'-ATTTGCTATTATGTAAGCCCC-3'). RT-PCR products were sent to Genoscreen (Lille, France) for sequencing. Sequences were visualized with the software ChromasPro v1.7.6 (www.technelysium.com.au/chromas.html) and aligned with the sequences of the PVY variants used to start the evolution experiment to detect mutations.

## Measures of virus replicative fitness and virulence

After the 7[th] infection cycle, all final populations resulting from the evolutionary lineages that did not become extinct and the three initial PVY variants were mechanically inoculated on the two cotyledons of the DH line on which they have evolved, with 20 plants per PVY population. Six mock-inoculated plants were also used as controls for each DH line. At 30 dpi, the virulence of the PVY populations was estimated by the impact of the virus on the height and the fresh weight of the plants (S3 Dataset). Fresh and dry weights of pepper plants belonging to the same progeny were shown previously to be highly correlated (r = 0.957, p < 0.0001) [38]. The plants were first cut at the cotyledon node and immediately measured and weighted. Then, 1 g from 3 apical leaves showing symptoms was sampled from each plant and a quantitative DAS-ELISA was performed to assess the final virus replicative fitness ($W_f$). $W_f$ and $W_i$ are expressed without unit relatively to a common PVY-infected plant sample. The gain of adaptation of the virus to the plant during the evolution experiment was then obtained with $\Delta W = W_f - W_i$.

## Measures of virus competitiveness

Competition experiments have been performed between the most frequent mutants of SON41-119N observed during the evolution experiment and the initial virus to assess their competitive replicative fitness. We used the SON41-119N variant and two variants derived by directed-mutagenesis from SON41p which carried either the 115K-119N or the 115M-119N pairs of amino acid substitutions in the VPg. Two competition experiments have been performed in the six DH lines HD219, HD2173, HD2256, HD2321, HD2344 and HD2349. For each DH line, 10 to 30 plants were inoculated with a 2:1 ratio mixture, based on quantitative DAS-ELISA, of the PVY variants SON41-119N and SON41-115K-119N, or SON41-119N and SON41-115M-119N. In each case, variant SON41-119N was the major component, because it was expected to have the lowest competitiveness based on evolution experiment results (Fig 4). Thirty days after inoculation, RNA extraction, RT-PCR of the VPg cistron and sequencing were realized as described above. To estimate the relative proportion of the PVY variants at the end of the competition experiment, the height of peaks at the codon position 115 were measured in the chromatograms. Although this method was shown to provide precise quantitative estimates of the relative proportions of the two virus variants in competition [39], we took into account only strong trends of virus frequency evolution. For this, plants were grouped in two categories where either (i) variant 1 or (ii) variant 2 predominated or was fixed in the population.

## Statistical analyses

All statistical analysis were performed with R software version 4.2.3 using the packages PMCMRplus (Dunnett and Nemenyi tests), car (VIF calculation), MuMIn (model selection),

pscl (McFadden's R-squared and R-squared), ResourceSelection (Hosmer-Lemeshow goodness-of-fit test) and visreg (for plotting glm results). Wilcoxon test was used to compare virus replicative fitness means between initial variant ($W_i$) and derived final populations ($W_f$). Kruskal–Wallis test followed by the Nemenyi *post hoc* test were used to compare virulence mean values between mock-inoculated plants and plants inoculated with either the initial PVY variants or the final PVY populations. Dunnett test was used to compare the virus replicative fitness as well as the virulence between every final PVY population and the corresponding initial PVY variants. For the competition experiments, chi-squared tests were used to compare the final frequencies of the two variants among all the plants belonging to the same DH line and the expected frequency of the two variants in the absence of competition (*i.e.* 2/3 of the plants where the single mutant SON41-119N was predominant or fixed, and 1/3 of the plants where the double mutants SON41-115K-119N or SON41-115M-119N was predominant or fixed). P-values were computed using Monte Carlo simulations (2,000,000 replicates) and Bonferroni correction was used to adjust for multiple comparisons.

Generalized linear models (GLM) were used to study the effects of $W_i$, $N_e$, and their two-way interactions on the evolutionary trajectories observed during the evolution experiment. Multiple $N_e$ estimates were available from a previous experiment [11]. We used $N_e$ estimated from 7 to 10 dpi which corresponds to the onset of plant systemic infection by PVY (see "Relevance and accuracy of variables used to explain the virus evolutionary trajectories" section for more details). Two response variables were tested: (i) the probability of PVY replicative fitness gain during the evolution experiment, which distinguishes the PVY lineages showing replicative fitness gain (n = 33 out of 64 lineages) or replicative fitness loss (n = 31 out of 64, including lineages that get extinct) after 7 infection cycles and (ii) the final replicative fitness $W_f$ of the 55 lineages that did not go to extinction during the experiment. The fit of the retained GLMs were assessed using the Hosmer and Lemeshow goodness of fit test and the chi-squared goodness of fit test for the probability of fitness gain and the final fitness $W_f$, respectively. We assumed a binomial distribution for the probability of replicative fitness gain and a Gaussian distribution for the final replicative fitness $W_f$. McFadden's R-squared and R-squared were also calculated for the probability of PVY replicative fitness gain and the final replicative fitness $W_f$, respectively. For both response variables, a stepwise model selection using the Akaike Information Criterion (AIC) was performed.

## Supporting information

**S1 Fig. Virulence of the final PVY populations that evolved on six pepper doubled-haploid (DH) lines.** Six plants per DH line were mock-inoculated (gray), while 20 plants per DH line were inoculated either with the variant SON41-119N (orange) or with each of the PVY populations derived from SON41-119N that were serially inoculated on these lines (blue). Boxplots of (A) plant height and (B) fresh weight at 30 days post inoculation are represented. The same experiment (C) was performed independently with the three initial variants SON41-119N, SON41-101G and SON41-115K as well as the resulting populations that were serially inoculated onto HD2173. Significance levels were obtained with the Kruskal–Wallis test followed by the Nemenyi *post hoc* test (ns: not significant, * p < 0.05, ** p < 0.01).
(TIF)

**S2 Fig. Virulence of the eight final PVY populations inoculated serially onto the pepper lines HD219 and HD2344.** The initial variant SON41-119N (orange) and each of the final PVY populations (blue) were inoculated onto 20 plants. Boxplots of plant height (A) and fresh weight (B) at 30 days post inoculation are represented. The letters a and b indicate the different groups obtained after the comparison of each final PVY population to the initial variant using

Dunnett test ($p < 0.05$).
(TIF)

**S1 Table. Traits characterizing each pepper DH line (virus effective population size $N_e$) or DH line–Initial PVY variant combination (virus initial replicative fitness $W_i$) used for the evolution experiment.** When possible, the broad-sense heritability ($h^2$) of the trait among plant DH lines was calculated.
(DOCX)

**S2 Table. Competition experiments between PVY variants in pepper DH lines.** The inoculum was composed of a 2:1 ratio mixture of the two variants in competition, the single mutant SON41-119N being the most concentrated. Thirty days after inoculation, the relative proportions of each variant were estimated by sequencing. Plants were grouped in two categories where either the single (SON41-119N) or double mutant (either SON41-115K-119N or SON41-115M-119N) predominated or was fixed in the population. The plant segregation according to the predominant virus was compared to the frequencies expected in the absence of selection (2:1 ratio of single *vs*. double PVY mutant) with a chi-squared test and Bonferroni correction for multiple testing (* $p < 0.05$, *** $p < 0.001$).
(DOCX)

**S3 Table. Comparison of virus variant distributions in the plants after one month of competition.** Plants belonging to six DH lines were inoculated with a 2:1 ratio mixture of the PVY single mutant SON41-119N and either the double mutant SON41-115K-119N or the double mutant SON41-115M-119N. One month after inoculation, the composition of the virus population in each plant was assessed by sequencing. Plants were grouped in two categories where either (i) SON41-119N or (ii) the double mutant predominated or was fixed in the population. The table displays the non-corrected p-values obtained after performing chi-squared tests to compare the final frequencies of the two variants between the DH lines. The final frequencies were also compared to the null hypothesis (H0), which states that the expected frequencies of the two variants in the absence of competition should correspond to 2/3 of plants where the single mutant SON41-119N is predominant or fixed, and 1/3 of plants where the double mutants SON41-115K-119N or SON41-115M-119N is predominant or fixed. Bonferroni correction was used to correct for multiple testing (* $p < 0.05$, ** $p < 0.01$, *** $p < 0.001$). Significant p-values according to Bonferroni correction are highlighted in grey.
(DOCX)

**S1 Dataset. A CSV file containing replicative fitness values measured for each plant.** This file serves as one of three input files for the R Markdown script analyzing the data (S1 File).
(CSV)

**S2 Dataset. A CSV file containing both explanatory and response variables used in the GLMs.** This file is one of three input files for the R Markdown script analyzing the data (S1 File).
(CSV)

**S3 Dataset. A CSV file containing virulence values measured for each plant.** This file is one of three input files for the R Markdown script analyzing the data (S1 File).
(CSV)

**S1 File. R Markdown script encompassing all statistical analyses carried out in this article, along with the generated figures and tables.**
(RMD)

**S2 File. HTML-formatted R Markdown report containing comprehensive statistical analyses conducted in this article, along with the generated figures and tables.**
(HTML)

## Acknowledgments

This study was performed with the LBM platform and the INRAE experimental facilities of the Plant Pathology research unit (https://eng-pathologie-vegetale.paca.hub.inrae.fr/infrastructures/prophyle/experimental-facilities). We thank Guillaume Martin (ISEM, Montpellier, France) for helpful discussions on our results.

## Author Contributions

**Conceptualization:** Lucie Tamisier, Frédéric Fabre, Alain Palloix, Benoît Moury.

**Data curation:** Lucie Tamisier, Frédéric Fabre.

**Formal analysis:** Lucie Tamisier, Frédéric Fabre, Benoît Moury.

**Funding acquisition:** Alain Palloix, Benoît Moury.

**Investigation:** Lucie Tamisier, Marion Szadkowski, Lola Chateau, Ghislaine Nemouchi, Grégory Girardot, Pauline Millot, Alain Palloix, Benoît Moury.

**Project administration:** Alain Palloix, Benoît Moury.

**Validation:** Lucie Tamisier, Frédéric Fabre.

**Visualization:** Lucie Tamisier, Frédéric Fabre.

**Writing – original draft:** Lucie Tamisier, Frédéric Fabre, Benoît Moury.

**Writing – review & editing:** Lucie Tamisier, Frédéric Fabre, Lola Chateau, Benoît Moury.

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
