## [Decision Letter · Decision Letter 0]

23 Oct 2023

Dear Dr. Tamisier,

Thank you very much for submitting your manuscript "Within-plant genetic drift to control virus adaptation" for consideration at PLOS Pathogens. As with all papers reviewed by the journal, your manuscript was reviewed by members of the editorial board and by several independent reviewers. In light of the reviews (see below), we would like to invite the resubmission of a significantly-revised version that takes into account the reviewers' comments.

Three reviewers all agree in the interest of the study and in the significant effort devoted by the authors to improve the original version on the manuscript. Still, some concerns remain, which are particularly related to the measures of Ne and s, and to their interpretation. Although no new experiments are suggested, some significant revision will be necessary to address these comments. Thus, we invite you to respond to the reviewer(s)' comments and revise your manuscript.

We cannot make any decision about publication until we have seen the revised manuscript and your response to the reviewers' comments. Your revised manuscript is also likely to be sent to reviewers for further evaluation.

Sincerely,

Israel Pagán

Guest Editor

PLOS Pathogens

Shou-Wei Ding

Section Editor

PLOS Pathogens

Kasturi Haldar

Editor-in-Chief

PLOS Pathogens

orcid.org/0000-0001-5065-158X

Michael Malim

Editor-in-Chief

PLOS Pathogens

orcid.org/0000-0002-7699-2064

Reviewer's Responses to Questions

**Part I - Summary**

Reviewer #1: The authors performed experimental evolution using Potato virus Y (PVY) on different pepper lines. The main aim of the study was to investigate whether virus adaptation could be controlled by favorably breeding certain plant genotypes over others. More specifically the authors asked the question whether “manipulation” of selection and genetic drift could help control virus adaptation and prevent a resistance breakdown. The question asked and the results presented are relevant to the field. However, the study also has some major weaknesses, as already pointed out by previous Reviewers #1 and #2. Most of these weaknesses could/were not addressed experimentally, but the authors have provided additional explanations to their results and acknowledged the shortcomings of their methods in the manuscript. These changes were crucial for the reader to better understand the study. Nevertheless, I also agree that the implementation of manipulating selection and genetic drift in the wild is quite a long shot.

Reviewer #2: I have now read the manuscript “Within-plant genetic drift to control virus adaptation”. In this revised version, the authors have integrated a large part of the comments made by the reviewers of the first round of revisions and the quality and clarity of the manuscript has improved substantially. The question tackled in interesting and timely. The experiments are well conducted.

Reviewer #3: Plant viruses pose a significant threat to crop yields. To mitigate these losses, there have been efforts to introduce genes that confer resistance to these viruses. However, these viruses can evolve to overcome such resistance. The authors have employed experimental evolution in an exceptionally well-designed experiment to assess how host-imposed conditions affect virus evolution. They discovered that host genotypes enforcing low effective population size (Ne) and reduced replicative fitness in the virus hinder the virus's ability to adapt to the host, evidenced by numerous virus lineages going extinct under these conditions.

While the authors primarily frame their findings within an agricultural context, the implications extend beyond this scope. The insights are also valuable for understanding virus evolution more broadly.

The manuscript has undergone prior review before my evaluation. The preceding reviewers have conducted a thorough examination of this manuscript and have raised pertinent comments. Based on the authors' responses and the revised manuscript, it appears that the authors have satisfactorily addressed these comments; the manuscript now includes more detailed and/or clarified text in response to reviewers' suggestions, enhancing both the manuscript's comprehensibility and the precision of its discussions.

**Part II – Major Issues: Key Experiments Required for Acceptance**

Reviewer #1: 1. The authors promote that the best solution to an increased resistance durability is to use plant genotypes/lines on which the virus has a low effective population size (Ne) and a low initial viral replicative fitness (Wi) However, in nature it is unlikely that only one virus variant circulates, and therefore it is most likely hard to find a suitable plant genotype for all viral variants. Indeed their results demonstrate that the outcome can be highly variable depending on the pepper line and virus variant used, where a high Ne and a high Wi appears to also prevent virus adaptation (pepper HD2173 and virus SON41-115K). The same accounts for the combination of a high Ne and intermediate Wi (pepper HD2173 and virus SON41 119N).

2. In a previous study the authors had already estimated the selection coefficient (s) and the effective population size (Ne) of PVY on the pepper lines used in this study. As also pointed out by previous Reviewer #2, these estimates were done using a more complex mixture of PVY variants (5 to be precise). Whereas in this study 3 of these variants have been evolved in isolation in different pepper lines. This can have a large effect on the s and Ne estimates. In their answer to previous Reviewer #2, the authors explain they have performed additional experiments to measure Ne with a more direct method (Ne_inoc instead of Ne_systemic). Even though this method may also not be an ideal way to measure Ne (the presence of GFP might also have an effect on viral replication success), I think it is a valuable addition to the manuscript. Therefore, it is not clear to me why the authors decided not to include the results of this additional measurement in the manuscript/supplementary material.

3. There is only information about Ne for the first passage. Ne_systemic and Ne_inoc were both estimated from 7 to 10 days post infection. There is no information on Ne in the following passages. As only a very small sample of the plant was taken for the following passages, I assume that the bottleneck size was quite strong, selecting only for those variants that are present at a high frequency in the virus population. It is important to consider this effect as well.

Reviewer #2: The point that remains problematic is the use of the variables Ne and s, derived from experimental data in a previous study by the same group and published in PLoS Pathogens. Regarding Ne, I find problematic that the authors use the Ne evaluated from 7 to 10 dpi to explain evolutionary dynamics occurring in an experimental set up where viruses are passaged every 28 days. It means that they account for only part of the story. In particular, the infection bottleneck is likely to have a quite important role and it is not accounted for. This should at least be discussed and ideally a more time-integrative measure of Ne should be used.

Regarding s, I find it very difficult to understand what s represents. It seems to be the opportunity for selection among 5 viral variants in a plant genotype. First, I find the name “s” confusing as it is usually used for the selective coefficient characteristic of a variant (in competition with a reference strain) and not of a host or environment. Then it is measured as “the difference between the growth rates of the fittest and the weakest variants” and the identity of these two variants is not the same from one host genotype to another and also potentially different from the two variants in competition in each of the experimentally evolving lines (i.e. the variant used to start experimental evolution and the main variant arising by mutation in this lineage), such that the potential explanatory link between this “s” and the evolution of a resistance breakdown is not clear from the beginning. And this variable actually never has a significant effect in the models presented, which is actually not surprising. My suggestion is actually to remove totally this variable from the analysis, as both the introduction and the interpretation are focus of Ne and drift effects. It would make the whole story a lot clearer and more straightforward.

Reviewer #3: There are two issues with the wording that need clarification or revision for the manuscript to be considered accepted (persuasive explanations may also suffice):

L236-237 "Combining a strong within-plant genetic drift and a low PVY initial replicative fitness can prevent virus evolution"

The authors note the emergence of new mutations and alterations in replicative fitness. They found this even without investigating mutations in other genes, shifts in the virus quasispecies, or different virus fitness phenotypes. Therefore, asserting that virus evolution is entirely prevented is misleading (is it possible to entirely halt evolution?). I strongly suggest rephrasing this to "Combining a strong within-plant genetic drift and a low PVY initial replicative fitness can prevent virus adaptation to host-resistance genes" or similar.

Similarly, I propose refining the title of the manuscript for greater specificity, perhaps to "Within-plant genetic drift to control virus adaptation to host-resistance genes", since the authors cannot confirm if the viruses are undergoing other forms of adaptation.

**Part III – Minor Issues: Editorial and Data Presentation Modifications**

Reviewer #1: 1. Something that is unclear: If the authors went through the effort of doing a 7 month long evolution experiment, why didn’t they use Illumina sequencing to sequence the entire genome of the ancestral and evolved virus populations? I am not asking the authors to do this. But with the low sequencing costs nowadays, especially of a small viral genome, this only makes sense to me. This would have given a more quantitative estimation of the variants present in the VPg cistron, show what happens in the rest of the genome, and possibly give explanations of fitness gains/losses that can not be explained with the current data.

2. Make all raw data/scripts available (Zenodo repository) so that the analyses can be reproduced.

3. Along the whole manuscript correct phrases like in line 214: “..that were fixed during the experimental evolution..” to “..that were fixed during the evolution experiment..”

4. Line 455: correct the figure number: “..experimental evolution results(Figs 2A, 2B ..)”

Reviewer #2: Minor comments:

L311-315 + l333: Muller’s ratchet can indeed explain extinctions but here, the authors haven’t detected any mutations in the populations which got extinct, so the accumulation of deleterious mutations do not seem to be the reason why they got extinct. A purely demographic effect seems more likely.

In their response to one of the previous reviews the authors write “we argue that pyramiding major resistance gene(s) with QTLs imposing a strong genetic drift within a plant would best reduce virus adaptation and would be preferable in terms of durability potential”. The authors actually do not say this in the paper. The conclusion remains very vague on the strategies to reduce Ne and thus adaptation potential and would strongly benefit from adding this more “mechanistic” strategy.

L177: no lack of adaptive mutations but can there be a difference in mutation supply (linked to the number of viral replications)?

Typos and formulation:

L25: repetition of « contrasted ».

L27: I guess the authors mean “linking” and not “inking”.

L29: replace “evolutionary forces” by “drift and selection”

L33: replace “and constant whatever” by “independently of”

L105: “divergent” is not the best word choice in this context as it suggests divergent evolution. “different” might be better.

L112: clarify “initial viral load”: is it the viral load at x dpi for the ancestral population or the viral load just after inoculation?

L117: remove “and” at the end of the line

L121: adaptation to what? To which plant genotype?

L185-187: phrase mal foutue

L259: final replicative fitness remains is high independently of the replicative fitness of the initially inoculated virus.

L344: remove ‘viral”

L345: “favouring” instead of “favoured”? Otherwise I do not get this sentence.

L594: Rousseau et al is reference 11 and not 10

Reviewer #3: 1) The are numerous studies showing how in different potyviruses the VPg is where more adaptive mutations are accumulated. I suggest to add some of these references to strength the decision of only sequencing the VPg gene. This could go after L126.

Furthermore, the authors might consider citing instances of potyviruses adapting to hosts through mutations in other genomic regions and explicitly acknowledging that mutations in the non-sequenced portions of the genome could contribute to their findings. For instance, the statement in L177 about the “lack of adaptive mutations for PVY in these plant genotypes” should be more concise and refer only to l” “lack of adaptive mutations for VPg protein of PVY TO t

---

## [Decision Letter · Decision Letter 1]

1 Jul 2024

Dear Dr. Tamisier,

We are pleased to inform you that your manuscript 'Within-plant genetic drift to control virus adaptation to host-resistance genes' has been provisionally accepted for publication in PLOS Pathogens.

Best regards,

Israel Pagán

Guest Editor

PLOS Pathogens

Shou-Wei Ding

Section Editor

PLOS Pathogens

Michael Malim

Editor-in-Chief

PLOS Pathogens

orcid.org/0000-0002-7699-2064

Dear Dr. Tamisier,

The reviewers have now gone through the revised version of your manuscript and both agree in that all their previous concerns have been addressed. Thus, I am happy to accept the manuscript for publication. Please, for your final version of the workpay attention to the minor editing that Reviewer 2 suggests.

Congratilations on the publication of this excellent work.

Best regards,

Reviewer Comments (if any, and for reference):

Reviewer's Responses to Questions

**Part I - Summary**

Reviewer #2: The revised version of the manuscript is much clearer. All my concerns have been addressed and the correspoding changes in the manuscript have been made.

Reviewer #3: The authors have addressed all my concerns and suggestions satisfactorily. From my side, the manuscript is now suitable for publication.

**Part II – Major Issues: Key Experiments Required for Acceptance**

Reviewer #2: (No Response)

Reviewer #3: (No Response)

**Part III – Minor Issues: Editorial and Data Presentation Modifications**

Reviewer #2: There are some problems with figure numbering:

- figure 2 disappeared in this version and is not referred to, so I understand it was a decision to remove it. I think it was quite useful but the manuscript can also stand without it. But the following figures haven't been renumbered and this has to be done.

- figure 5 disappeared in this version but is still referred to, so I guess it has been forgoten when submiting the new version pf the manuscript. This has to be corrected.

References 14 and 15 are identical.

Reviewer #3: (No Response)

PLOS authors have the option to publish the peer review history of their article (what does this mean?). If published, this will include your full peer review and any attached files.

Reviewer #2: No

Reviewer #3: No

---

## [Editor Report · Acceptance letter]

31 Jul 2024

Dear Dr. Tamisier,

We are delighted to inform you that your manuscript, "Within-plant genetic drift to control virus adaptation to host-resistance genes," has been formally accepted for publication in PLOS Pathogens.

Best regards,

Michael Malim

Editor-in-Chief

PLOS Pathogens

orcid.org/0000-0002-7699-2064